# Solution-processed intermediate-band solar cells with lead sulfide quantum dots and lead halide perovskites

Hiroji Hosokawa [1], Ryo Tamaki[2], Takuya Sawada[1], Akinori Okonogi[1], Haruyuki Sato[1], Yuhei Ogomi[3], Shuzi Hayase [3], Yoshitaka Okada [2] & Toshihiro Yano[1]

The intermediate-band solar cell (IBSC) with quantum dots and a bulk semiconductor matrix has potential for high power conversion efficiency, exceeding the Shockley-Queisser limit. However, the IBSCs reported to date have been fabricated only by dry process and their efficiencies are limited, because their photo-absorption layers have low particle density of quantum dots, defects due to lattice strain, and low bandgap energy of bulk semiconductors. Here we present solution-processed IBSCs containing photo-absorption layers where lead sulfide quantum dots are densely dispersed in methylammonium lead bromide perovskite matrices with a high bandgap energy of 2.3 eV under undistorted conditions. We confirm that the present IBSCs exhibit two-step photon absorption via intermediate-band at room temperature by inter-subband photocurrent spectroscopy.

[1] Material Science Research, R&D, Kao Corporation, 1334 Minato, Wakayama 640-8580, Japan. [2] Research Center for Advanced Science and Technology, The University of Tokyo, 4-6-1 Komaba, Meguro-ku, Tokyo 153-8904, Japan. [3] Graduate School of Life Science and Systems Engineering, Kyushu Institute of Technology, 2-4 Hibikino, Wakamatsu-ku, Kitakyushu, Fukuoka 808-0196, Japan. Correspondence and requests for materials should be addressed to H.H. (email: hosokawa.hiroji@kao.com)

ntermediate-band solar cells (IBSCs) have strong potential for high power conversion efficiency (PCE), exceeding the Shockley-Queisser limit[1]. The IBSC has an intermediate-band (IB) between the valence band (VB) and conduction band (CB), within the bandgap of bulk semiconductor in a photo-absorption layer[2–4]. The IBSC provides two-step photon absorption (TSPA) from the VB to IB and from the IB to CB, in addition to the bulk absorption from the VB to CB. Quantum dots (QDs) incorporated into the bulk matrix have been considered as one of the most promising candidates to realize the IBSCs[5–11]. Presently, such IBSCs have been fabricated only by dry process such as the molecular beam epitaxy (MBE) and metal organic chemical vapor deposition (MOCVD). However, it is generally difficult to improve the PCE in dry-processed IBSCs, because restrictions on the materials that can be used make it difficult to increase in the particle density of QDs and in the bandgap energy ($E_{BG}$) of bulk in the photo-absorption layer of the IBSC[3,4,7,9,10]. Also, the dry-processed IBSCs contain photo-absorption layers with lattice strain introduced by a large lattice-mismatch between the QDs and the bulk matrix, resulting in a decrease in PCE[3,4,6,7,10] due to strained-induced defects. In contrast, a solution process has been proposed as a means to fabricate the IBSC. The colloidal QDs have been proposed to be a promising platform for the IBSC[12]. However, no experimental reports on solution-processed IBSC with QDs have been reported to date. Here, we create IBSCs containing photo-absorption layers where QDs are densely dispersed in a bulk matrix with a high $E_{BG}$ under undistorted conditions by the solution process. The IBSCs are confirmed to exhibit TSPA via IB at room temperature by inter-subband photocurrent spectroscopy.

## Results

**Design and fabrication.** Taking into account the solution process, we have selected a perovskite compound[13–19] as the bulk material and the colloidal QD[20–24] as the QD-IB material. Lead (Pb) halide perovskite and lead sulfide (PbS) QDs were chosen from the viewpoint of lattice-matching[25–27]. Based on the theoretical calculation of IBSC with a single IB under no concentration (one sun), the optimized band structure has an $E_{BG}$ of the bulk of 2.4 eV, which is split by the IB into two sub-bandgaps of 1.5 eV and 0.9 eV[4]. The IB corresponds to the CB of QDs[3,4]. Taking these energy levels into consideration, we selected methylammonium lead bromide ($CH_3NH_3PbBr_3$) perovskite, having $E_{BG}$ of 2.3 eV[14], as well as PbS QDs with the size of 4 nm, having $E_{BG}$ of 1.0 eV[25], which gives an energy difference between the VB maximum of the bulk and IB ($E_{VI}$) of 1.5 eV and energy difference between the IB and CB minimum of the bulk ($E_{IC}$) of 0.8 eV (Fig. 1a). In order to obtain a photo-absorption layer with the QD particle density of not less than $1 \times 10^{12}$ cm$^{-2}$ per layer which is the highest density achieved by the MBE method[28], the PbS QDs of 11 volume% or more were hybridized with the perovskite matrix (Fig. 1b).

Uniform films of $CH_3NH_3PbBr_3$ perovskite can be prepared by spin-coating from a mixture solvent of γ-butyrolactone (GBL) and dimethyl sulfoxide (DMSO)[14]. However, the photo-absorption layers where PbS QDs are dispersed in the perovskite matrix could not be fabricated from the mixture solvent of GBL and DMSO, because PbS QDs had low dispersibility in the mixture solvent. Consequently, in the present study, we use N, N-dimethylformamide (DMF) solvent, in which iodine-capped PbS QDs can well disperse[23]. The photo-absorption layers are prepared by spin-coating from DMF dispersions containing iodine-capped PbS QDs with the size of 4 nm and perovskite raw materials (PbBr$_2$, CH$_3$NH$_3$Br) (Methods).

**Structural characterization.** Figure 2 shows X-ray diffraction (XRD) patterns of the photo-absorption layers on mesoporous TiO$_2$ (mTiO$_2$) layers. The hybridization of the PbS QDs with the perovskite did not change the XRD peak positions of the perovskite. No peak due to PbS at $2\theta = 25$ degree was detected. The XRD results revealed that $CH_3NH_3PbBr_3$ perovskite with undistorted lattice[29] was the main component of the photo-absorption layers.

Scanning electron microscopy (SEM) observations of the photo-absorption layer surface (Fig. 3) indicate the presence of cubic perovskite crystals with the size of 100 to 200 nm. Although the hybridization of the PbS QDs led to coexistence of the smaller perovskite particles with the size of a few-tens nm, the surface coverage of the perovskite on the substrates (70 to 80%) did not depend on the amounts of the PbS QDs.

Based on the cross-sectional SEM image (Fig. 4a), the thickness of the photo-absorption layer with the PbS QDs of 14 volume% was between 100 nm to 200 nm. In the magnified SEM image of the photo-absorption layer (Fig. 4b), inside the gray perovskite region, spherical particles (white regions) of 3 to 15 nm in particle size appear in a scattered manner, indicating how the PbS QDs have been scattered within the perovskite matrix[25,26]. Furthermore, high-resolution transmission electron microscopy (HRTEM) observations (Fig. 4c) show how spherical particles (black regions), with the size of 3 to 15 nm, were densely dispersed, although the larger particles were formed by aggregation of the PbS QDs with the size of 4 nm by electron beams. In addition, when magnifying and observing the PbS QDs (black regions, Fig. 4d) and the perovskites (gray region, Fig. 4e), lattice fringes matching the interplanar spacing (0.34 nm) of the (111) plane of cubic PbS[30] and the interplanar spacing (0.30 nm) of the (200) plane of tetragonal $CH_3NH_3PbBr_3$[29] are seen, confirming that the PbS QDs were dispersed in the $CH_3NH_3PbBr_3$ perovskite matrix under the undistorted condition.

**Optical properties.** No change in the perovskite absorption peak of 530 nm was observed even after hybridizing the PbS QDs (Fig. 5), indicating that the perovskite $E_{BG}$ of 2.3 eV was constant. Near-infrared (NIR) photoluminescence (PL) spectra excited at 785 nm exhibited emission peaks of the PbS QDs at 1210 nm (Fig. 5). Although the emission intensity increased with increasing content of the PbS QDs, the peak wavelength remained constant. Thus, the $E_{BG}$ of PbS QDs in the photo-absorption layers was determined to be 1.0 eV. It has been reported that PbS QDs with $E_{BG}$ of 1.0 eV have a mean particle diameter of 4 nm[25].

Note that NIR emission of the PbS QDs was observed even by exciting the perovskite (excitation wavelength of 532 nm, Supplementary Fig. 1a) in the photo-absorption layers. This result indicates that a part of the charged-carriers photogenerated in the perovskite transfers to the PbS QDs[25–27], due to the fact that the VB maximum and the CB minimum of PbS QDs exist within the bandgap of the perovskite (Fig. 1a). The carrier transfer from the perovskite to PbS QDs was also supported by experimental results that emission intensity and lifetime of the perovskite decreased with the small amounts of the PbS QDs (Supplementary Fig. 1b, c), while the decrease in the emission intensity and lifetime of the perovskite may be partly attributed from the lower crystallinity of the perovskite (Supplementary Fig. 2).

As the excitation power density increased, the NIR emission intensity linearly increased and the emission peak wavelength blue-shifted (Fig. 6, Supplementary Fig. 3), confirming electron coupling between the PbS QDs, i.e., the IB formation[31] in the photo-absorption layers with the PbS QDs and the perovskite. The emission peak shift was nearly saturated at about 1.058 eV when

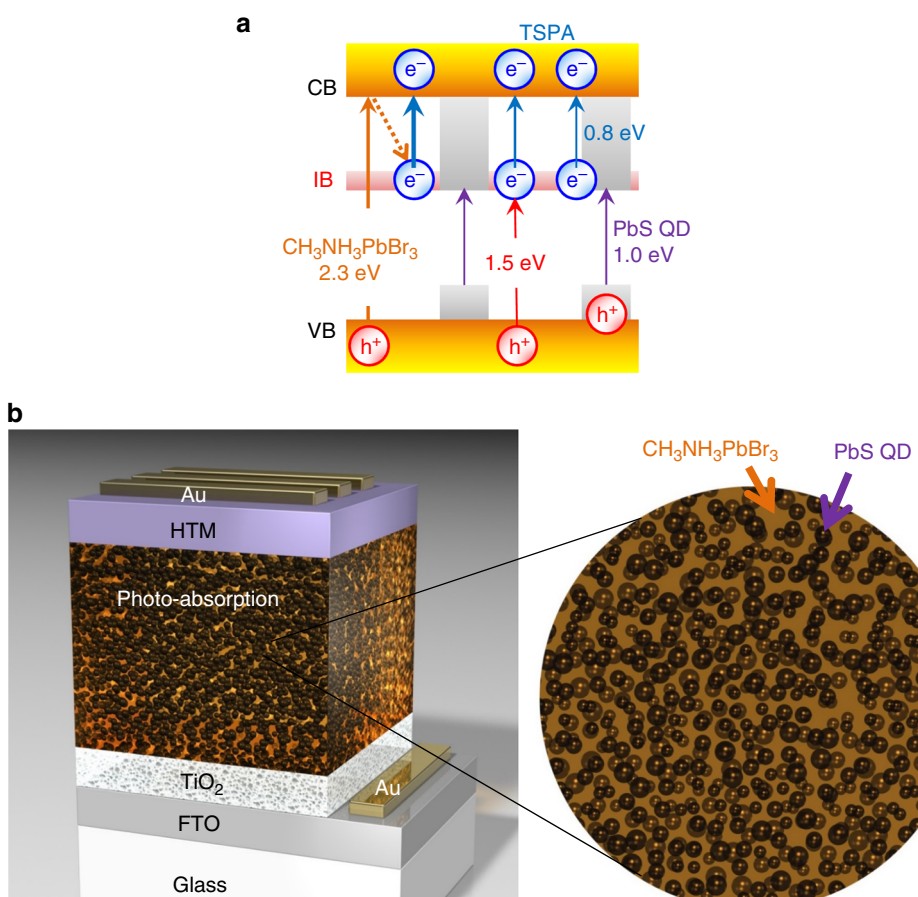

**Fig. 1** Design of solution-processed IBSC with PbS QDs and $CH_3NH_3PbBr_3$ perovskite. **a** Schematic energy band diagram of the photo-absorption layer. $E_{BG}$ of $CH_3NH_3PbBr_3$ perovskite; 2.3 eV, $E_{BG}$ of PbS QD; 1.0 eV, $E_{VI}$; 1.5 eV, $E_{IC}$; 0.8 eV. **b** Structural model of the IBSC. The photo-absorption layer, where PbS QDs with the mean size of 4 nm are densely dispersed in the perovskite matrix, is placed between $TiO_2$ and hole-transporting material (HTM)

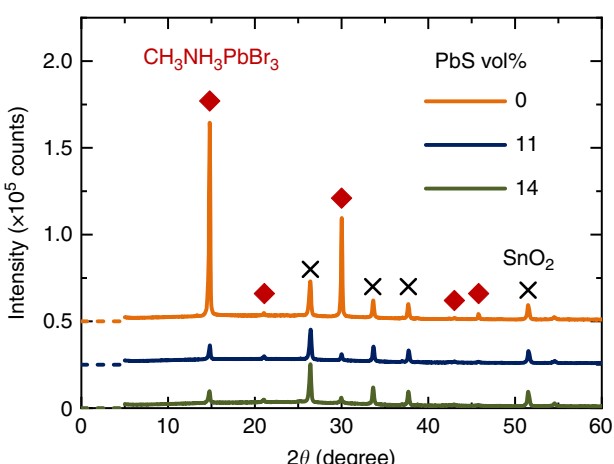

**Fig. 2** XRD patterns of the photo-absorption layers. The peaks at $2\theta = 14.8$, 21.1, 30.0, 43.0, 45.8 degree correspond to the (100), (110), (200), (220), (300) reflections of cubic $CH_3NH_3PbBr_3$ perovskite. The peaks at $2\theta = 26.4$, 33.6, 37.7, 51.5 degree assign to $SnO_2$ of FTO substrates

excited at the excitation density of 1226 W cm$^{-2}$. The blue-shifted energy was calculated to be 0.02 eV, which corresponds to the width of their IB. In summary, the band structure of the photo-absorption layer prepared was identified as designed (Fig. 1a).

**Solar cell properties.** External quantum efficiency (EQE) values in the visible light region from 400 nm to 550 nm (Fig. 7a) were decreased by hybridizing the PbS QDs, while EQE values in NIR light region from 600 nm to 1250 nm (Fig. 7b) were increased with the PbS QDs. The EQE onset wavelengths (550 nm and 1250 nm) were consistent with $E_{BG}$ of the perovskite (2.3 eV) and the PbS QDs (1.0 eV), respectively. Thus, the EQE values in the visible and NIR light regions are derived from the charged-carriers photogenerated in the perovskite and the PbS QDs, respectively. Consequently, the decrease of EQE in the visible region for the cells with PbS QDs should be explained in terms of the carrier transfer from the perovskite to the PbS QDs as described above. As a result, PCE values were decreased by hybridizing the PbS QDs (Supplementary Fig. 4). The changes in the solar cell properties by the PbS QDs are similar to those by InAs QDs in dry-processed IBSC. However, the NIR EQE due to the PbS QDs (0.1 to 0.2% at 1000 nm, Fig. 7b) was much lower than that in the dry-processed IBSC (6% at 920 nm)[8], suggesting that thermal excitation from the IB to the perovskite CB may be suppressed because of the larger $E_{IC}$ (0.8 eV) in the present IBSCs.

The values of ΔEQE using IR bias light with more than 1319 nm (less than 0.94 eV) increased with the PbS QDs (Fig. 7c, d). In the visible light region (Fig. 7c), ΔEQE spectra steeply increased at 550 nm which was consistent with $E_{BG}$ of the perovskite (2.3 eV). Consequently the ΔEQE spectra between 500 nm and 550 nm are derived from the photo-excitation of the perovskite.

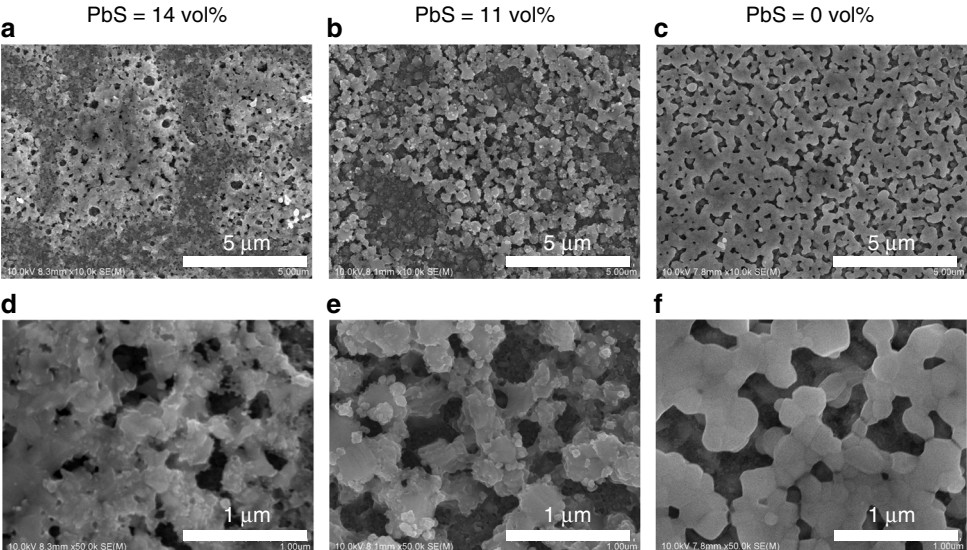

**Fig. 3** Top-view SEM images of the photo-absorption layers. The PbS volume concentrations in the photo-absorption layers were 14 volume% (**a**, **d**), 11 volume% (**b**, **e**), and 0 volume% (**c**, **f**), respectively

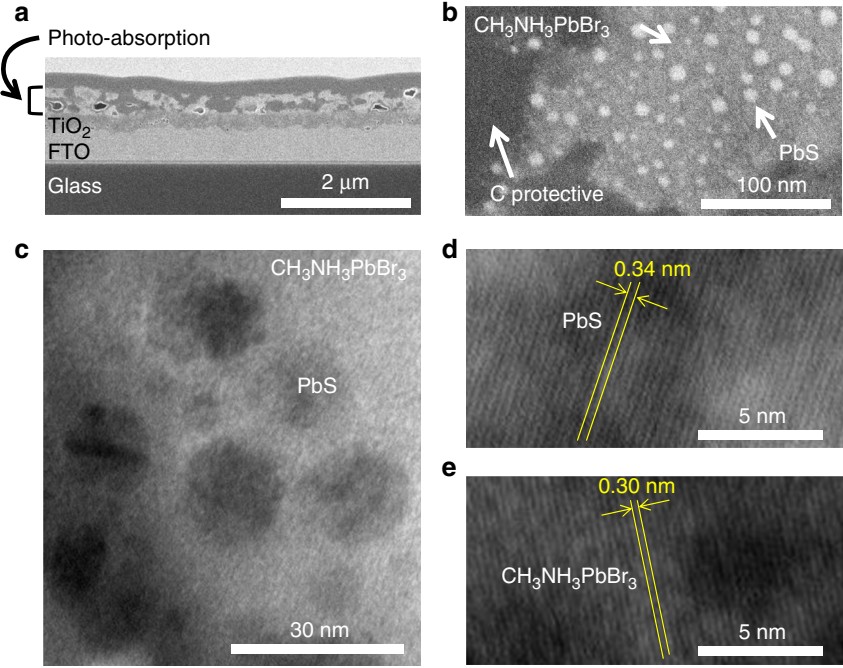

**Fig. 4** Cross-sectional SEM and HRTEM images. **a** Cross-sectional SEM image. From the bottom, glass substrate, FTO layer, TiO$_2$ layer, photo-absorption layer, carbon protective layer, tungsten protective layer. **b** High-magnification SEM and **c** HRTEM images of the photo-absorption layer. **d** HRTEM image of the PbS QD (black part). **e** HRTEM image of the CH$_3$NH$_3$PbBr$_3$ perovskite (gray part)

Taking into account the fact that the perovskite cell without the PbS QDs has no IB, the ΔEQE spectrum of the perovskite cell, which was almost independent on the IR bias light energy (Supplementary Fig. 5, 6) should arise from the photo-excitation of shallow-trapped electrons on defects of the perovskite[32]. In contrast, the ΔEQE values of the hybrid cells with the PbS QDs fell by the decrease in IR bias light energy from less than 0.94 eV to less than 0.83 eV (Supplementary Fig. 5, 6), indicating the narrow energy distribution of the IB (0.02 eV, Fig. 6) with the $E_{IC}$ of 0.8 eV (Fig. 1a). Thus, the ΔEQE spectra between 500 nm and 550 nm of the hybrid cells with the PbS QDs should be explained in terms of the TSPA of the IB electrons transferred from the

perovskite CB, since the carrier transfer from the perovskite to the PbS QDs[25–27] has been supported by the emission behavior (Supplementary Fig. 1) and the cell characteristics (Fig. 7a, Supplementary Fig. 4). In the NIR light region (Fig. 7d), the hybrid cells with the PbS QDs exhibited ΔEQE spectra with the onset at around 1250 nm and the rising at around 850 nm in contrast to the perovskite cell with ΔEQE = 0%. The wavelengths of the onset and the rising were consistent with the $E_{BG}$ of PbS QDs (1.0 eV) and the $E_{VI}$ (1.5 eV), respectively (Fig. 1a). Consequently, the ΔEQE spectra between around 850 nm and around 1250 nm can be attributed to the TSPA where electrons photo-excited from the PbS QD VB to the IB (PbS QD CB) are

further photo-excited to the perovskite CB. The ΔEQE spectra ranging from 600 nm to around 850 nm should be partly due to the TSPA of the IB electrons photo-excited from the perovskite VB. Thus, the hybrid cells with the PbS QDs exhibited the TSPA via the IB, confirming that they were functioning as the IBSCs at room temperature.

## Discussion

We prepared the photo-absorption layers by the solution process of spin-coating from the DMF dispersions containing the PbS QDs with the size of 4 nm and the perovskite raw materials ($PbBr_2$, $CH_3NH_3Br$). The structural and spectroscopic characterizations of the photo-absorption layers indicated that the PbS QDs with the $E_{BG}$ of 1.0 eV were densely dispersed in $CH_3NH_3PbBr_3$ perovskite matrices with the $E_{BG}$ of 2.3 eV under undistorted conditions as designed in Fig. 1. The blue-shift in NIR PL with increasing excitation density (Fig. 6b) confirmed IB formation in the photo-absorption layers. The cells with the photo-absorption layers exhibited the TSPA via the IB at room temperature by the inter-subband photocurrent spectroscopy as shown in Fig. 7c, d. Thus, we can conclude that a solution process was a powerful tool to fabricate IBSCs without limitations often reported for dry-processed IBSCs.

As shown in Supplementary Fig. 4, the PCE values of the present IBSCs with the PbS QDs and the perovskite were low because of some factors such as the carrier transfer from the perovskite to the PbS QDs and poor film quality. However, an incorporation of a tunneling barrier with CB minimum above that of the perovskite into the interface between the PbS QDs and the perovskite may block the carrier transfer from the perovskite to the PbS QDs, leading to hign PCE[33]. And also, the carrier transfer may be suppressed by control of their energy levels like type-II structure[4] which can be realized by hybridizing a QD material with VB maximum below that of the perovskite. In addition, the TSPA process should be enhanced by increasing the lifetime of carriers in the intermediate state with photon ratchet[34] and in type-II structure[4]. Further investigations are currently underway to enhance the solar cell performance by suppressing the carrier transfer and by improving the film quality. The present results pave the way towards not only high PCE but also low cost, flexible IBSCs.

## Methods

**Synthesis of PbS quantum dots (QDs).** Oleate-capped PbS quantum dots (QDs) were synthesized according to a previously published method[35]. A mixture of 0.45 g of lead oxide (99.999%), 10 g of 1-octadecene (ODE, more than 95%), and 1.34 g of oleic acid (more than 90%) was degassed at 353 K for 2 h. The obtained solution was heated to 383 K and kept for 30 min under $N_2$, followed by rapid injection of an ODE solution (4 mL) of 1,1,1,3,3,3-hexamethyldisilathiane (0.21 mL). After injection, the obtained colloidal solution was allowed to cool down to room temperature, and PbS QD solid was separated by adding acetone and centrifugation. The oleate-capped PbS QDs were characterized by inductively coupled plasma (ICP) analysis, X-ray photoelectron spectroscopy (XPS), proton nuclear magnetic resonance ($^1$H NMR) spectroscopy, X-ray diffraction (XRD), absorption and photoluminescence (PL) spectroscopies, as described below. The weight concentrations of Pb and oleic acid in the oleate-capped PbS QD solid were measured to be 55 wt% and 28 wt% by ICP analysis and $^1$H NMR spectroscopy, respectively. The molar ratio of oleate to Pb (oleate/Pb) could be calculated to be 0.37. The atomic ratio of Pb/S/I/N was determined to be 1/0.58/0/0 by using XPS. The crystalline size of the oleate-capped PbS QDs was measured to be 3.0 nm by XRD. The bandgap energy ($E_{BG}$) of the oleate-capped PbS QDs was estimated to be 1.2 eV from absorption onset wavelength (1050 nm) and emission peak wavelength (1040 nm) excited at 800 nm.

The above oleate-capped PbS QD solid could be dispersed in toluene, but not in N,N-dimethylformamide (DMF), which was a good solvent for a solution of perovskite raw materials ($PbBr_2$, $CH_3NH_3Br$). Consequently iodine (I)-capped PbS QDs with high dispersibility in DMF were synthesized by ligand exchange process[23] at room temperature. In the ligand exchange process, DMF-solvated I-ligands replace oleate ligands on the PbS QD surface. In a glovebox, 0.20 g of the above oleate-capped PbS QD solid was dispersed in 2 mL of toluene (super dehydrated). A mixed solution of 1 mL of toluene, 0.5 mL of DMF (super dehydrated), and 0.062 g of $CH_3NH_3I$ (MAI) was dropwise added to the PbS dispersion for 11 min without stirring. The molar ratio of MAI to oleate of PbS QDs (MAI/oleate) could be calculated to be 2. After 18 h, 5 mL of methanol (super dehydrated) was added to the PbS dispersion in order to precipitate I-capped PbS QD solid. The I-capped PbS QD solid was separated by filtration using polytetrafluoroethylene (PTFE) filters with the pore size of 0.20 μm. The weight

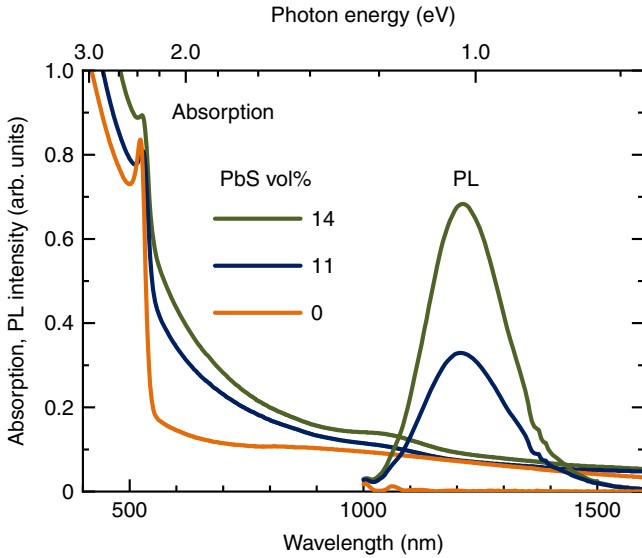

**Fig. 5** Absorption and NIR PL spectra of the photo-absorption layers at room temperature. Excitation wavelength was 785 nm

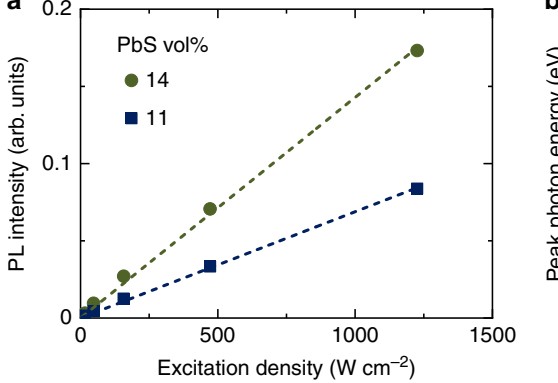
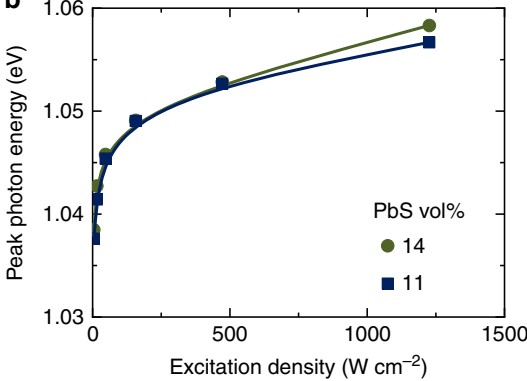

**Fig. 6** Excitation density dependence of NIR PL of the photo-absorption layers. **a** Emission intensity and **b** emission peak energy at room temperature. Excitation wavelength was 532 nm

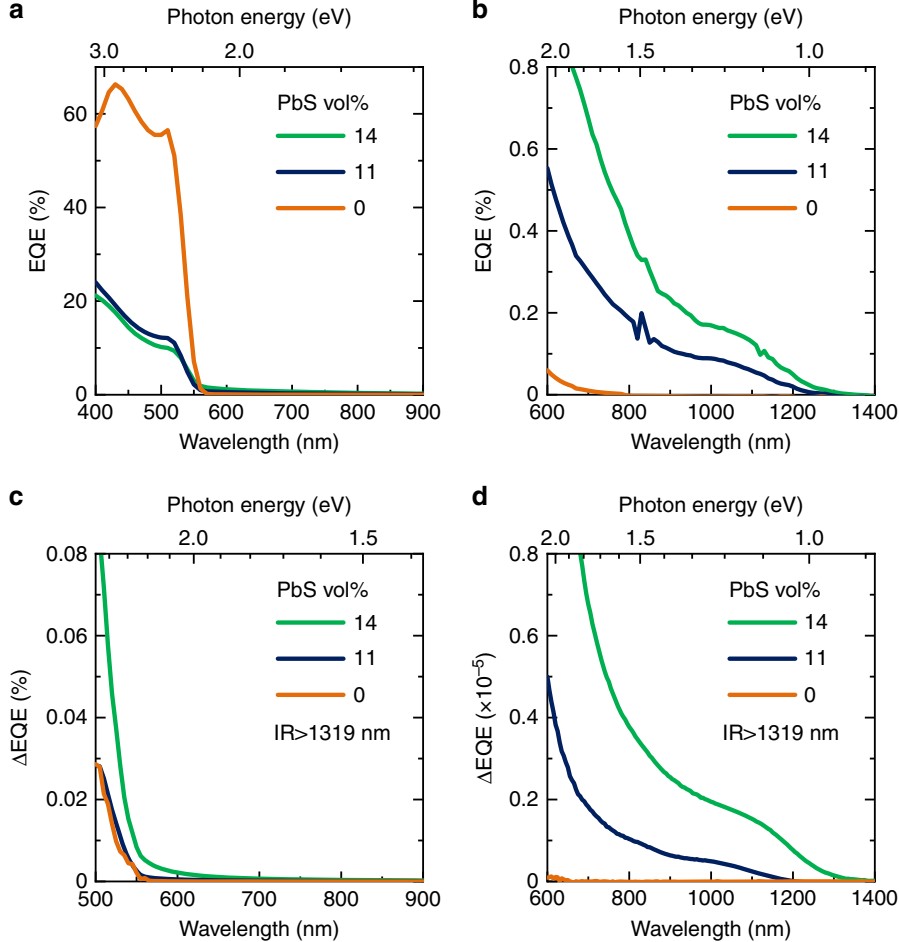

**Fig. 7** Spectral response of solution-processed IBSCs with PbS QDs and $CH_3NH_3PbBr_3$ perovskite. **a** EQE, **b** magnified EQE, **c** ΔEQE, and **d** magnified ΔEQE spectra at room temperature. The wavelength of IR bias light was more than 1319 nm

concentrations of Pb and oleic acid in the I-capped PbS QD solid were measured to be 55 wt% and 1 wt% by ICP analysis and [1]H NMR spectroscopy, respectively. The molar ratio of oleate to Pb (oleate/Pb) could be calculated to be 0.01. The atomic ratio of Pb/S/I/N was determined to be 1/0.51/0.49/0 by using XPS. The crystalline size of the I-capped PbS QDs was measured to be 3.5 nm by XRD. The $E_{BG}$ of the I-capped PbS QDs was determined to be 1.0 eV from absorption onset wavelength (1200 nm).

**Solar cell fabrication**. A compact blocking $TiO_2$ (c$TiO_2$) layer with the thickness of less than 30 nm was deposited on a UV-ozone-cleaned fluorine-doped tin oxide (FTO)-coated glass substrate (Asahi Glass Co., Ltd., 25 × 25 × 1.8 mm) by spray pyrolysis at 723 K from a precursor solution of 4.04 g bis(isopropoxide)bis(acet-ylacetonato)-titanium(IV) (75 wt% in 2-propanol) in 123.24 g ethanol (super dehydrated). The c$TiO_2$ layer was treated with 0.05 mol L$^{-1}$ aqueous solution of $TiCl_4$ at 343 K for 30 min and rinsed with water. The c$TiO_2$ substrate was sintered at 773 K for 20 min. A mesoporous $TiO_2$ (m$TiO_2$) layer with the thickness of about 200 nm was coated on the c$TiO_2$ substrate by spin-coating with a speed of 5000 rpm for 30 s from a diluted solution of 20 nm $TiO_2$ particle paste (PST-18NR, JGC Catalysts and Chemicals Ltd.) with the weight ratio of $TiO_2$ paste: ethanol of 1: 3.5, and then the substrate was sintered at 773 K for 30 min.

As described above, we used DMF solvent for the perovskite precursor ($PbBr_2$, $CH_3NH_3Br$), because the above I-capped PbS QDs could well disperse in DMF[23]. In the preliminary experiment in DMF solvent, we performed the precursor concentration (0.15 to 1.0 mol L$^{-1}$) studies on the solar cell properties of the perovskite cells without PbS QDs though not shown here. From the viewpoint of their solar cell properties, the precursor concentration in DMF was selected to be 0.31 mol L$^{-1}$. In the glovebox, the above I-capped PbS QD solids (0 to 0.20 g) were mixed with 0.31 mol L$^{-1}$ solutions (2 mL) of $PbBr_2$ (perovskite precursor grade) and $CH_3NH_3Br$ in DMF (super dehydrated) with stirring at room temperature for 15 min, and then the resulting solutions were filtered with PTFE filters with the pore size of 0.45 μm. The PbS weight concentrations (mg mL$^{-1}$) in the filtrates were determined from absorbance (A) at the peak wavelength (960 to 1100 nm)

after appropriate dilution by DMF (PbS = 1.2 mg mL$^{-1}$ A$^{-1}$). The PbS volume concentrations in the photo-absorption layers (volume% = (volume of PbS)/ {(volume of PbS) + (volume of $CH_3NH_3PbBr_3$)} × 100) were calculated on the basis of the density of 7.5 g cm$^{-3}$ for PbS and that of 3.8 g cm$^{-3}$ for $CH_3NH_3PbBr_3$[36]. The photo-absorption layer was deposited on the m$TiO_2$ substrate by spin-coating with speeds of 500 rpm for 5 s, 1000 rpm for 40 s, and 3000 rpm for 50 s with the respective slopes of 5 s from the filtrates. One milliliter toluene (super dehydrated) as a poor solvent was dripped onto the substrate center at 20 s after the beginning of the spin-coating. After that, the photo-absorption layer on the m$TiO_2$ substrate was dried on a hot plate at 373 K for 10 min. At the same time, the photo-absorption layer was similarly fabricated on a glass substrate (Matsunami Glass Ind., Ltd., 26 × 25 × 1.0 mm) as described above for measurement of their optical properties. We verified that the photo-absorption layer on the glass substrate was almost equivalent to that on the m$TiO_2$ substrate by means of XRD, absorption spectroscopy, and SEM observation.

The hole-transporting material (HTM) layer was deposited on the photo-absorption layer by spin-coating with a speed of 4000 rpm for 30 s from a 0.058 mol L$^{-1}$ solution of (2,2′,7,7′-tetrakis(N,N-di-p-methoxyphenylamine)-9,9-spirobifluorene) (Spiro-OMeTAD) in chlorobenzene containing 4-tert-butylpyridine (0.19 mol L$^{-1}$), lithium bis-(trifluoromethylsulfonyl)imide (0.031 mol L$^{-1}$), and tris(2-(1H-pyrazol-1-yl)-4-tert-butylpyridine) cobalt(III) (5.6 mmol L$^{-1}$). The resulting film was dried on a hot plate at 343 K for 30 min.

Finally, 100 nm of gold was thermally evaporated with the deposition rate of 0.8 to 0.9 nm s$^{-1}$ on top of the HTM layer under high vacuum (4 to 5 × 10$^{-3}$ Pa).

**Structural characterization**. XRD patterns were recorded on a X-ray dif-fractometer (Rigaku, Mini Flex600, light source CuKα, tube voltage 40 kV, tube current 15 mA) under the conditions of sampling width of 0.02 degree, scan speed of 20 degree min$^{-1}$, solar slit of 5.0 degree, divergence slit of 13.0 mm, scan range of 2θ = 5 to 60 degree. The crystalline size of the PbS QDs was determined at cubic PbS (220) peak (2θ = 42 degree) based on Scherrer equation by using analysis software (PDXL ver.2.6 1.2).

The weight concentrations of Pb in the PbS QD solids were determined by ICP analysis of clear solutions dissolved in HNO$_3$/H$_2$O$_2$ mixture[37]. The weight concentrations of oleic acid in the PbS QD solids were measured by $^1$H NMR spectra with dibromomethane as an internal standard in deuterated toluene (99 atom%D, tetramethylsilane (TMS) 0.03 volume%) or deuterated DMF (99.5 atom%D)[38]. The $^1$H NMR data were obtained on a NMR spectrometer (Agilent, VNMR400), operating at $^1$H frequency of 400 MHz, delay time of 60 s, integration of 32 times. The atomic ratios of Pb/S/I/N in the PbS QD solids were determined from the area ratios of Pb 4 f, S 2p, I 3d, and N 1 s in XPS data[23,39]. The XPS spectra were measured on a XPS spectrometer (Ulvac-phi, PHI Quantera SXM) using monochromatic AlKα radiation (25 W, 15 kV) under the conditions of beam size of 100 μm, area of 1 mm$^2$, pulse energy of 112 eV, step of 0.2 eV, detection angle of 45 degree, binding energy correction C1s (284.8 eV).

Scanning electron microscopy (SEM) images of the photo-absorption layer surface were obtained on a field emission-scanning electron microscope (FE-SEM, Hitachi, S-4800), operating at 10 kV. The fraction of surface coverage of the perovskite on the mTiO$_2$ layer was calculated from the SEM images (magnification of 10,000) by using an image analysis software (WinROOF, Mitani Corp.). The structure of the PbS QDs and the perovskite in the photo-absorption layer with the PbS QDs of 14 volume% was analyzed at 153 K using a cryo-system for less perturbation of electron beams. The cross-sectional film samples with the thickness of 150 nm were prepared at 153 K by slicing on a micro-sampling focused ion-beam apparatus (FIB, Hitachi, NB-5000) after depositions of carbon (C) and tungsten (W) as protective layers. The cryo-SEM images (secondary electron) were obtained on a scanning and transmission electron microscope (STEM, Hitachi, HD-2300), operating at 200 kV. High-resolution transmission electron microscope (HRTEM) characterization was performed at 153 K using high-resolution transmission electron microscope (Hitachi, H-9500), operating at 300 kV. Under the STEM and HRTEM observations, we found that some particles of the PbS QDs were aggregated to form the larger particles with the size of more than 10 nm by the electron beams. Thus, we could not exactly determine the particle size and the particle density of the PbS QDs in the photo-absorption layer by STEM and HRTEM. Although the lattice constants of PbS[30] and MAPbBr$_3$[29] are well matched, we could not get direct evidence of the match of the lattice orientation[25] between PbS QDs and MAPbBr$_3$ matrix.

**Optical properties**. Transmission absorption spectra of dispersions of the PbS QDs and of the photo-absorption layers on glass substrates were measured at room temperature on an ultraviolet (UV)-visible (Vis)-near infrared (NIR) spectrometer (Shimadzu, Solid Spec-3700) under the conditions of a detective unit of integrating sphere, scanning speed of normal, sampling pitch of 1 nm, slit width of 20, measurement range of 300 to 1600 nm. Background measurements were performed using toluene or DMF solvent in a quartz cell (10 mm path length) for the PbS QD dispersions and the glass substrate for the photo-absorption layers on glass substrates, respectively. In the transmission absorption spectra of the photo-absorption layers, no absorption peak at around 1000 nm assigned to the PbS QDs was observed, so it was not possible to estimate the $E_{BG}$ of the PbS QDs in the photo-absorption layers.

NIR PL spectra ranging from 1000 to 1600 nm of the photo-absorption layers on glass substrates were measured at room temperature on a photoluminescence spectrometer using laser diode light source (15 mW) at excitation wavelength of 532 nm (excitation of the perovskite) and 785 nm (excitation of the PbS QDs). Excitation density dependence of the NIR PL spectra was performed at room temperature under the conditions of excitation wavelength of 532 nm, excitation power from 1 to 78 mW, and laser spot diameter of 90 μm. The integrated PL emission intensity and PL emission peak wavelengths were determined by Gaussian-fitting. Visible light PL spectra ranging from 500 to 850 nm were obtained at room temperature on a photoluminescence spectrometer (Horiba, Fluorolog) under the conditions of excitation wavelength of 480 nm, slit width of excitation light of 1 nm, slit width of emission light of 5 nm, dark-offset on. Time-resolved PL experiments were performed at room temperature on a photoluminescence lifetime measurement apparatus (Hamamatsu Photonics, Quantaurus-Tau C11369). Emission decay curves from 0 to 100 ns of the perovskite in the photo-absorption layers on glass substrates were measured at 544 nm with a time resolution of 100 ps (excitation wavelength 470 nm). The emission lifetimes were calculated on the basis of two-exponential-decay fitting.

**Solar cell properties**. The photocurrent density-voltage (J-V) curves of the cells with the photo-absorption layers were recorded at room temperature in air using a solar simulator (Peccell Technologies, PEC-L01) under AM 1.5 G condition (100 mW cm$^{-2}$). Cell active area of 0.036 cm$^2$ was defined by a black metal mask. Scan rate, step voltage, search delay, hold time, and scan range were fixed at 0.1 V s$^{-1}$, 0.01 V, 0.05 s, 0.05 s, and −0.1 V to 1.1 V, respectively. The light intensity was corrected with a calibrated Si reference cell (Bunkoukeiki, BS-520).

The external quantum efficiency (EQE) spectra of the cells with the photo-absorption layers were measured at room temperature in air on a spectral response measurement system (Bunkoukeiki, CEP-2000MLR, direct current (DC) method) with a metal mask to give an active area of 0.036 cm$^2$. The power of the incident monochromatic light was kept under 2.5 mW cm$^{-2}$, which was calibrated with a Si reference cell (Bunkoukeiki, BS-520BK).

The difference in EQE (ΔEQE) spectra between with and without IR bias light illumination were measured at room temperature in air using two-step photon absorption (TSPA) photocurrent spectroscopy with IR bias light[4,5,7–11]. The wavelength of the IR bias light was more than 1319 nm or 1500 nm. The IR bias light can pump electrons only from the intermediate-band (IB) to conduction band (CB) of the perovskite neither from the valence band (VB) to IB nor from the VB to CB. The ΔEQE spectra were obtained on a spectral response measurement setup (alternating current (AC) method, active area 0.036 cm$^2$) by using a lock-in amplifier synchronized with an optical chopper set at 5 Hz. A halogen lamp (100 W) was used as a monochromatic light source with the photon density ranging from $1 \times 10^{13}$ photon cm$^{-2}$ s$^{-1}$ at 500 nm to $1 \times 10^{16}$ photon cm$^{-2}$ s$^{-1}$ at 1100 nm, which was enough low to ignore filling of the IB. The IR bias light from the other tungsten lamp passed through the optical chopper and an appropriate set of filters that allowed only the IR region of more than 1319 nm or more than 1500 nm to be transmitted. The power of the IR bias light was 56 mW cm$^{-2}$ (more than 1319 nm) or 50 mW cm$^{-2}$ (more than 1500 nm). Difference spectra in the ΔEQE were calculated based on subtraction the ΔEQE using the long-pass filter of 1500 nm (ΔEQE$_{1500}$) from the ΔEQE using the long-pass filter of 1319 nm (ΔEQE$_{1319}$).

## Data availability

All relevant data are available from the corresponding author upon reasonable request.

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

## Acknowledgements

We acknowledge Dr. A. Wakamiya at Kyoto University for his advice in the fabrication of solar cells. A part of this work was conducted at Kyushu University, supported by Nanotechnology Platform Program (Molecule and Material Synthesis) of the Ministry of Education, Culture, Sports, Science and Technology (MEXT), Japan.

## Author contributions

H.H., T.S., H.S., and T.Y. conceived the design. H.H., R.T., T.S., A.O., and Y.Og. carried out the experiments and analyzed the results. H.H. and R.T. co-wrote the manuscript with input from all authors. S.H., Y.Ok., and T.Y. supervised the project.

## Additional information

**Competing interests:** The authors declare no competing interests.

