## [Peer Review File · Nature Communications]

Reviewers' comments:

Reviewer #1 (Remarks to the Author):

The manuscript contributed by Hosokawa et al. reports the intermediate-band solar cells (IBSC) using PbS quantum dot (QD)/perovskite hybrid material as the active layer. IBSC is potentially useful to achieve high power conversion efficiency. The observation of the wavelength-dependent EQE change under IR bias is also important in relation to the deepened understanding of the photophysical behaviors of QD/perovskite hybrids. However, I am not convinced that the QD/MAPbBr₃ hybrid reported in the current manuscript is potentially good for the solar cell application – instead of giving better charge separation function, the formation of the type-I structure may actually cause the trapping of free charge carriers. More importantly, some characterization results hint that the incorporation of QD lowers the quality of perovskite crystals. The introduction of introduces new recombination sites at the interface between QD and perovskites may be the reason why the PCE is lower when QDs are incorporated. Because of these reasons, I do not recommend this work be accepted by Nature Communications in the current form.

More detailed comments are listed below:

1. The key question in this study is about whether the mixture of PbS QD and MAPbBr₃ brings any benefits for solar cell application. Since the bandgap of PbS is much smaller than that of MAPbBr₃ perovskite (also shown in Figure 1), it is clear that the incorporation of PbS forms type-I structure and creates new recombination centers for free charge carriers. This idea has been applied for highly efficient light-emitting diodes (X. Gong et al., Nature Photonics, 2017, 10, 253).

However, the efficient trapping of free charge carriers within the active layer is undesired here since it makes the separation and transport of free charge carriers become more difficult and thus impair the performance of solar cells. This is indirectly supported by the device performance results shown in Figure S2, where lower current density and PCE values are reported.

2. Instead of forming a uniform and smooth thin film, pinholes are clearly shown in all samples (perovskite, QD/perovskite hybrid) in Figure 3. These SEM results indicate that the recipes for the formation of perovskite and QD/perovskite hybrid are not optimized well for the fabrication of solar cell absorbing layer. Indeed, the PL lifetime decay of QD-free MAPbBr₃ thin film (5.6 ns) is much faster than many published numbers, suggesting the defect density is already high in QD-free MAPbBr₃ thin film. On the other hand, I am not quite sure if the thickness of the absorbing layer has been optimized – in perovskite solar cell society, it is widely accepted that a thick active layer (~500 – 600 nm) is important for efficient light absorption and conversion. However, the active layer used in this study is much thinner. I would therefore recommend the authors carry out thickness-dependent or precursor concentration studies for improving the quality of the active material.

3. The coherence of the lattice orientation is the key to improve surface passivation of PbS QD by perovskites (Z. Ning, et al., Nature, 2015, 523, 324). Although the lattice constants of PbS and MAPbBr₃ are well matched (<1% lattice mismatch), I do not find any evidence that confirms the match of the lattice orientation between QDs and the perovskite matrix (at least this is not clearly shown in Figure 4).

4. Some characterization results hint that the incorporation of QD lowers the quality of perovskite crystals. The intensity of the crystalline signals from perovskites decreased dramatically when QDs are introduced (Figure 2). Considering only a small amount of QDs (<15 vol%) are present, ~10x dropping of crystalline signal intensity is abnormal. The imperfect crystallization may also cause the

faster PL decay shown in Figure S1, although the authors consider the shorter lifetime is because of the fast charge transfer from perovskite to QDs.

The poor crystallinity of perovskite (when QDs are incorporated) suggests that the dynamic of crystalline formation of perovskite is changed when QD is introduced during the film processing. This is also reflected by the aggregation of the QDs present in the hybrid film (Figure 4). I notice the authors used iodide-passivated QDs, whereas PbI₃-passivated QDs are used in other reports (e.g., Z. Ning, et al., *Nature*, 2015, 523, 324; Z. Yang, et al., *Nano Lett.*, 2015, 15, 7539; X. Zhang, et al., *Adv. Energy. Mater.*, 2018, 8, 1702049). I was wondering if the low quality of perovskite crystalline and aggregation of QDs are because of the interaction between the surface ligands of QDs and perovskite precursors are different than those reported in the literature.

Reviewer #2 (Remarks to the Author):

The manuscript by Hosokawa et al. brings an original approach to the intermediate band solar cell (IBSC) concept. The use of colloidal quantum dots might contribute to solve the bottleneck that present QD IBSCs show in terms of absorption due to their higher density that seems that can be achieved with this approach. Although the results are not outstanding in terms of solar cell performance, in my view, they, together with the discussions by the authors, point to the right direction. Because of this I recommend the manuscript is published.

Reviewer #3 (Remarks to the Author):

(1) The starting proposition to make Intermediate Band Solar Cells (IBSCs) from Quantum Dots (QDs) is a very profound one. The IBSC and the QD fields are both huge, but their synergy has not been even started or contemplated yet.

(2) Using PbSe perovskite QDs is a very intriguing idea, as perovskite QDs hold the world record of QD SC efficiency.

(3) The structural characterization is extensive and very impressive. The optical characterization is very good too.

(4) The Introduction should describe the history more comprehensively. To our knowledge, the QD-IBSC idea was first proposed theoretically by Voros, Galli and Zimanyi in 2015, see *ACS Nano*, 2015, 9 (7), pp 6882–6890.

(5) A central issue with IBSCs is the competition of time scales: the upconverting TSPA process is competing with the photo-induced electrons at the bottom of the perovskite CB decaying into the IB. This latter process of course is damaging the EQE. It would be very nice if the authors could provide a quantitative description of these two corresponding time scales and show that the upconversion rates beat the decay rates. This may be possible from lineshape studies.

(6) In the original 1997 Marti-Luque IB paper, the width and the filling of the IB play important roles. The IBSCs are supposed to be most efficient if the IB is narrow and half filled. I ask the authors to characterize the width and filling of their IB. I am a bit anxious about the width of the IB, since it comes from the CB of the QD which is quite wide.

(7) The overall effect on the EQE is a reduction for wavelengths below 550nm, and a small enhancement for wavelengths above 550nm. This is similar to what epitaxial ("dry method") IBSC papers report. Please provide a quantitative comparison: is the EQE enhancement bigger in IBSCs with QD or with dry epitaxial methods.

(8) Smaller remark: Please rephrase the sentence on p. 5 "larger particles were formed by perturbation of electron beams". In its present form this sentence is quite unclear. Maybe what was meant that QDs coalesced/aggregated into larger QDs?

In sum: I think the basic proposition of the paper, the QD-IBSC, needs to be heard in high profile publications. Also, the characterization and the results are comprehensive and compelling. Therefore, I strongly support the publication of this paper after the authors address the points (4)-(8).

Responses to Reviewers' Comments

Reviewer #1

The manuscript contributed by Hosokawa et al. reports the intermediate-band solar cells (IBSC) using PbS quantum dot (QD)/perovskite hybrid material as the active layer. IBSC is potentially useful to achieve high power conversion efficiency. The observation of the wavelength-dependent EQE change under IR bias is also important in relation to the deepened understanding of the photophysical behaviors of QD/perovskite hybrids. However, I am not convinced that the QD/MAPbBr₃ hybrid reported in the current manuscript is potentially good for the solar cell application – instead of giving better charge separation function, the formation of the type-I structure may actually cause the trapping of free charge carriers. More importantly, some characterization results hint that the incorporation of QD lowers the quality of perovskite crystals. The introduction of introduces new recombination sites at the interface between QD and perovskites may be the reason why the PCE is lower when QDs are incorporated. Because of these reasons, I do not recommend this work be accepted by Nature Communications in the current form.

More detailed comments are listed below:

1. The key question in this study is about whether the mixture of PbS QD and MAPbBr₃ brings any benefits for solar cell application. Since the bandgap of PbS

is much smaller than that of MAPbBr₃ perovskite (also shown in Figure 1), it is clear that the incorporation of PbS forms type-I structure and creates new recombination centers for free charge carriers. This idea has been applied for highly efficient light-emitting diodes (X. Gong et al., Nature Photonics, 2017, 10, 253).

However, the efficient trapping of free charge carriers within the active layer is undesired here since it makes the separation and transport of free charge carriers become more difficult and thus impair the performance of solar cells. This is indirectly supported by the device performance results shown in Figure S2, where lower current density and PCE values are reported.

Reply:

We thank the reviewer for the comments regarding the issues of the IBSC with QDs and perovskites. The benefit of the mixture of PbS QDs and MAPbBr₃ for solar cell application is in the potential for enhancement in current density and PCE values by two-step photon absorption (TSPA). As the reviewer pointed out, the incorporation of PbS QDs led to the carrier transfer from MAPbBr₃ to PbS QDs, resulting in the lower current density and PCE values. However, the carrier transfer may be suppressed by control of their energy levels like type-II structure (T. Tayagaki et al., Appl. Phys. Lett., 108, 153901 (2016)). In addition, the TSPA process should be enhanced by increasing the lifetime of carriers in the intermediate state with photon ratchet (M. Yoshida et al., Appl. Phys. Lett., 100, 263902 (2012)). Indeed, preliminary results of time-resolved photocurrent spectroscopy on IBSCs in this study suggested existence of long-lived photo-carriers in PbS QDs. It could be interpreted as quick extraction of holes due to

thermal excitation at room temperature, and electrons remained at QDs suppressing radiative recombination. If photo-carrier separation of electrons and holes would happen, long-lived photo-carriers could contribute efficiently on TSPA process. As a result, high-performance IBSCs with QDs and perovskites are expected to be available.

We have added the following text in page 16 and References 26, 33, 34.

“As shown in Supplementary Figure 4, the PCE values of the present IBSCs with the PbS QDs and the perovskite were low because of some factors such as the carrier transfer from the perovskite to the PbS QDs and poor film quality. However, the carrier transfer may be suppressed by control of their energy levels like type-II structure³³. In addition, the TSPA process should be enhanced by increasing the lifetime of carriers in the intermediate state with photon ratchet³⁴. Further investigations are currently underway to enhance the solar cell performance by suppressing the carrier transfer and by improving the film quality.”

2. Instead of forming a uniform and smooth thin film, pinholes are clearly shown in all samples (perovskite, QD/perovskite hybrid) in Figure 3. These SEM results indicate that the recipes for the formation of perovskite and QD/perovskite hybrid are not optimized well for the fabrication of solar cell absorbing layer. Indeed, the PL lifetime decay of QD-free MAPbBr₃ thin film (5.6 ns) is much faster than many published numbers, suggesting the defect density is already high in QD-free MAPbBr₃ thin film. On the other hand, I am not quite sure if the thickness of the absorbing layer has been optimized – in perovskite solar cell society, it is widely accepted that a thick active layer (~500 – 600 nm) is important for efficient light absorption and conversion. However, the active layer used in this study is much

thinner. I would therefore recommend the authors carry out thickness-dependent or precursor concentration studies for improving the quality of the active material.

Reply:

We thank the reviewer for the valuable proposal. We agree with the comment that the authors carry out thickness-dependent or precursor concentration studies for improving the quality of the active material.

QD-free MAPbBr₃ films without the pinholes can be prepared by spin-coating from a mixture solvent of γ -butyrolactone (GBL) and dimethyl sulfoxide (DMSO), as reported in Ref. 14 (S. Ryu, et al., Energy Environ. Sci. 7, 2614 (2014)). In contrast, QDs/MAPbBr₃ hybrid films could not be formed from the mixture solvent of GBL and DMSO, because PbS QDs had low dispersibility in the mixture solvent. Consequently, in the present study, we used *N, N*-dimethylformamide (DMF) solvent, in which iodine-capped PbS QDs can well disperse, as reported in Ref. 23 (X. Lan, et al., Nano Lett. 16, 4630 (2016)).

As described below, we had already performed the precursor concentration studies on QD-free MAPbBr₃ cells but not on PbS QDs/MAPbBr₃ hybrid cells. The SEM observations of QD-free MAPbBr₃ films prepared from DMF solutions with various precursor concentrations (0.15 - 1.0 mol/L) revealed that the surface coverage (18 - 98 %) and thickness (200 – 400 nm) of MAPbBr₃ on mTiO₂ substrates increased as the precursor concentrations (0.15 - 1.0 mol/L) increased, as shown in Reply Figure 1, 2, 3A, 3B. On the other hand, by increasing in the precursor concentrations, PL lifetime of MAPbBr₃ films on glass substrates decreased (Reply Figure 3C), suggesting an increase in the defect density. As a result, maximum value of PCE of QD-free MAPbBr₃ cells

was at the precursor concentration of 0.31 mol/L (Reply Figure 3D, 4). In conclusion, the precursor concentration studies on QD-free MAPbBr₃ cells showed that the optimized precursor concentration in DMF solvent was the present one (0.31 mol/L) from the viewpoint of the solar cell properties, although pinholes existed in the MAPbBr₃ thin films (Fig. 3C). Thus, in the present study, we fabricated PbS QDs/MAPbBr₃ hybrid films by the spin-coating from DMF dispersions with the precursor concentration of 0.31 mol/L.

Reply Figure 1. Top-view SEM images of QD-free MAPbBr₃ photo-absorption layer. The precursor concentrations were 0.15 mol/L (A, D), 0.62 mol/L (B, E), and 1.0 mol/L (C, F). The surface coverage of MAPbBr₃ on mTiO₂ substrates increased with increasing in the precursor concentrations.

Reply Figure 2. Cross-sectional SEM images of QD-free MAPbBr₃ photo-absorption layer. The precursor concentrations were 0.15 mol/L (A), 0.31 mol/L (B), 0.62 mol/L (C), and 1.0 mol/L (D). The thickness of MAPbBr₃ photo-absorption layer on mTiO₂ substrates increased with increasing in the precursor concentrations.

Reply Figure 3. Precursor concentration dependence of QD-free MAPbBr₃. (A) surface coverage, (B) film thickness, (C) PL lifetime, and (D) PCE.

Precursor concentration (mol/L)	0.15	0.31	0.62	1.0
PbS (vol%)	0	0	0	0
PCE (%)	1.6	1.9	1.3	0.73
V_{oc} (V)	0.71	0.68	0.79	0.76
J_{sc} (mA/cm ²)	2.8	3.7	3.1	1.7
FF	0.78	0.75	0.55	0.56

Reply Figure 4. Photocurrent density-voltage (J-V) curves and photovoltaic parameters of QD-free MAPbBr₃ cells.

Furthermore, we studied on precursor concentration (thickness) dependence of PbS QDs/MAPbBr₃ hybrid films. The SEM observations confirm that the surface coverage (80 - 98 %) and thickness (200 – 500 nm) of the hybrid films on mTiO₂ substrates increased as the precursor concentrations (0.31 - 1.0 mol/L) increased, as shown in Reply Figure 5, 6. On the other hand, the higher precursor concentrations correspond to the higher concentrations of PbS QDs in the DMF solutions. The higher PbS concentrations should partly induce aggregation of PbS QDs in the DMF dispersions. Since the PbS aggregates should be removed by the filtration, the resulting filtrates had the lower PbS concentrations. As a result, the amounts of PbS QDs incorporated into the hybrid films decreased from 14 vol% to 8 vol% by increasing in the precursor concentrations from 0.31 mol/L to 1.0 mol/L. Under the condition of the precursor concentration of 0.62 mol/L, PbS QDs/MAPbBr₃ hybrid film with the equivalent PbS

concentration of 11 vol% and with the higher surface coverage (90 %) and the higher thickness (300 nm) could be prepared. So, we fabricated PbS QDs/MAPbBr₃ hybrid cells under the condition of the precursor concentration of 0.62 mol/L, and evaluated their solar cell properties. As in the case of QD-free MAPbBr₃ cells, PCE value of the hybrid cells prepared under the condition of the precursor concentration of 0.62 mol/L was less than that of 0.31 mol/L (Reply Figure 7). These results suggest that drastic changes in the recipes such as ligands of PbS QDs and solvent may be needed in order to enhance in PCE value of the hybrid cells by improving in the film quality.

Reply Figure 5. Top-view SEM images of PbS QDs/MAPbBr₃ hybrid photo-absorption layer. The precursor concentrations were 0.31 mol/L (A, D), 0.62 mol/L (B, E), and 1.0 mol/L (C, F). The surface coverage of MAPbBr₃ on mTiO₂ substrates increased with increasing the precursor concentrations.

Reply Figure 6. Cross-sectional SEM images of PbS QDs/MAPbBr₃ hybrid photo-absorption layer. The precursor concentrations and PbS concentrations were 0.31 mol/L and 14 volume% (A), 0.31 mol/L and 11 volume% (B), 0.62 mol/L and 11 volume% (C), and 1.0 mol/L and 8 volume% (D), respectively. The thickness of MAPbBr₃ photo-absorption layer on mTiO₂ substrates increased with increasing the precursor concentrations.

Precursor Concentration (mol/L)	0.62	0.31	0.31	0.31
PbS (vol%)	11	14	11	0
PCE (%)	0.28	0.55	0.39	1.9
V _{OC} (V)	0.36	0.38	0.37	0.68
J _{SC} (mA/cm ²)	2.5	3.0	2.3	3.7
FF	0.30	0.48	0.45	0.75

Reply Figure 7. Photocurrent density-voltage (J-V) curves and photovoltaic parameters of PbS QDs/MAPbBr₃ hybrid cells.

We have added the following texts in pages 3 and 19, respectively.

“Uniform films of $\text{CH}_3\text{NH}_3\text{PbBr}_3$ perovskite can be prepared by spin-coating from a mixture solvent of γ -butyrolactone (GBL) and dimethyl sulfoxide (DMSO)¹⁴. However, the photo-absorption layers where PbS QDs are dispersed in the perovskite matrix could not be fabricated from the mixture solvent of GBL and DMSO, because PbS QDs had low dispersibility in the mixture solvent. Consequently, in the present study, we used *N,N*-dimethylformamide (DMF) solvent, in which iodine-capped PbS QDs can well disperse²³.”

“As described above, we used DMF solvent for the perovskite precursor (PbBr_2 , $\text{CH}_3\text{NH}_3\text{Br}$), because the above I-capped PbS QDs could well disperse in DMF²³. In the preliminary experiment in DMF solvent, we performed the precursor concentration (0.15-1.0 mol/L) studies on the solar cell properties of the perovskite cells without PbS QDs though not shown here. From the viewpoint of their solar cell properties, the precursor concentration in DMF was selected to be 0.31 mol/L.”

3. The coherence of the lattice orientation is the key to improve surface passivation of PbS QD by perovskites (Z. Ning, et al., Nature, 2015, 523, 324). Although the lattice constants of PbS and MAPbBr_3 are well matched (<1% lattice mismatch), I do not find any evidence that confirms the match of the lattice orientation between QDs and the perovskite matrix (at least this is not clearly shown in Figure 4).

Reply:

As the reviewer pointed out, we could not get direct evidence of the match of the lattice orientation between PbS QDs and MAPbBr₃ matrix. In the previous report (Z. Ning, et al., Nature, 2015, 523, 324), MAPbI₃ matrix was formed with MAPbI₃-capped PbS QDs by the sequential deposition method (J. Burschka et al., Nature, 2013, 499, 316). In contrast, in the present study, MAPbBr₃ matrix was prepared with I-capped PbS by the fast deposition-crystallization method (M. Xiao, et al., Angew. Chem. Int. Ed. 2014, 53, 9898). We speculate that the faster crystallization rate of MAPbBr₃ and the different halogen atoms between MAPbBr₃ matrix (Br) and PbS surface ligand (I) may lead to inhibit epitaxial growth of MAPbBr₃ on the surface of I-capped PbS QDs, resulting in no evidence of the match of the lattice orientation.

We have added the following text in page 21.

“Although the lattice constants of PbS³⁰ and MAPbBr₃²⁹ are well matched, we could not get direct evidence of the match of the lattice orientation²⁵ between PbS QDs and MAPbBr₃ matrix.”

4. Some characterization results hint that the incorporation of QD lowers the quality of perovskite crystals. The intensity of the crystalline signals from perovskites decreased dramatically when QDs are introduced (Figure 2). Considering only a small amount of QDs (<15 vol%) are present, ~10x dropping of crystalline signal intensity is abnormal. The imperfect crystallization may also cause the faster PL decay shown in Figure S1, although the authors consider the shorter lifetime is because of the fast charge transfer from perovskite to QDs. The poor crystallinity of perovskite (when QDs are incorporated) suggests that the dynamic of crystalline formation of perovskite is changed when QD is introduced

during the film processing. This is also reflected by the aggregation of the QDs present in the hybrid film (Figure 4). I notice the authors used iodide-passivated QDs, whereas PbI₃-passivated QDs are used in other reports (e.g., Z. Ning, et al., Nature, 2015, 523, 324; Z. Yang, et al., Nano Lett., 2015, 15, 7539; X. Zhang, et al., Adv. Energy. Mater., 2018, 8, 1702049). I was wondering if the low quality of perovskite crystalline and aggregation of QDs are because of the interaction between the surface ligands of QDs and perovskite precursors are different than those reported in the literature.

Reply:

We thank the reviewer for the useful suggestions. We agree the comment that the imperfect crystallization may also cause the faster PL decay shown in Figure S1.

We have added XRD patterns in Supplementary Figure 2 and the following text in page 10.

“while the decrease in the emission intensity and lifetime of the perovskite may be partly attributed from the lower crystallinity of the perovskite (Supplementary Fig. 2).”

As the reviewer pointed out, the interaction between the surface ligands of QDs and perovskite precursors should play an important role in determining the quality of perovskite crystal and dispersibility of QDs. In the present study, we used I-passivated but not PbI₃-passivated PbS QDs because of MAPbBr₃ but not MAPbI₃ matrix. Further investigations are currently underway to improve the film quality by control of the interaction using Br- or PbI₃-passivated PbS QDs and by change in the solvent.

Reviewer #3

(1) The starting proposition to make Intermediate Band Solar Cells (IBSCs) from Quantum Dots (QDs) is a very profound one. The IBSC and the QD fields are both huge, but their synergy has not been even started or contemplated yet.

(2) Using PbSe perovskite QDs is a very intriguing idea, as perovskite QDs hold the world record of QD SC efficiency.

(3) The structural characterization is extensive and very impressive. The optical characterization is very good too.

Reply:

We thank the reviewer for these favorable evaluations (points (1)-(3)).

(4) The Introduction should describe the history more comprehensively. To our knowledge, the QD-IBSC idea was first proposed theoretically by Voros, Galli and Zimanyi in 2015, see ACS Nano, 2015, 9 (7), pp 6882–6890.

Reply:

We thank the reviewer for the valuable suggestion. We have added the following text in page 2 and the proposed report as References 12.

“In contrast, a solution process has been proposed as a means to fabricate the IBSC. The

colloidal QDs have been proposed to be a promising platform for the IBSC¹². However, no experimental reports on solution-processed IBSC with QDs have been reported to date.”

(5) A central issue with IBSCs is the competition of time scales: the upconverting TSPA process is competing with the photo-induced electrons at the bottom of the perovskite CB decaying into the IB. This latter process of course is damaging the EQE. It would be very nice if the authors could provide a quantitative description of these two corresponding time scales and show that the upconversion rates beat the decay rates. This may be possible from lineshape studies.

Reply:

We thank the reviewer for bringing up the central issue with IBSCs. In the present IBSCs, the lineshape studies could not be applied for the quantitative comparison between the TSPA and the decay rates because PbS QDs have size distribution. As shown in Figure 7A, the EQE values due to the carriers photogenerated in the perovskite were decreased by hybridizing the PbS QDs, suggesting that the TSPA process should be slower than the decay. The decay rates should be very fast (pico-second order) as estimated from the PL decay results (Supplementary Fig. 1C). Thus, prevention of the decay is one of the most important issues for enhancement in PCE value. The decay may be suppressed by optimizing in their energy levels like type-II structure (T. Tayagaki et al., Appl. Phys. Lett., 108, 153901 (2016)).

We have added the following text in page 16 and References 33, 34.

“As shown in Supplementary Figure 4, the PCE values of the present IBSCs with the PbS QDs and the perovskite were low because of some factors such as the carrier transfer from the perovskite to the PbS QDs and poor film quality. However, the carrier transfer may be suppressed by control of their energy levels like type-II structure³³. In addition, the TSPA process should be enhanced by increasing the lifetime of carriers in the intermediate state with photon ratchet³⁴. Further investigations are currently underway to enhance the solar cell performance by suppressing the carrier transfer and by improving the film quality.”

(6) In the original 1997 Marti-Luque IB paper, the width and the filling of the IB play important roles. The IBSCs are supposed to be most efficient if the IB is narrow and half filled. I ask the authors to characterize the width and filling of their IB. I am a bit anxious about the width of the IB, since it comes from the CB of the QD which is quite wide.

Reply:

We thank the reviewer for giving us an opportunity to characterize the width and filling of their IB. We determined the width of their IB by excitation density dependence of photoluminescence (PL) peak energy as shown in Reply Figure 9, 10. It is worth noting that PL intensity linearly increased and PL peak wavelength blue-shifted as the excitation density increased. The PL peak shift was nearly saturated at about 1.058 eV when excited at the excitation density of 1226 W/cm². The blue-shifted energy was calculated to be 0.02 eV, which corresponds to the width of their IB. We estimated

almost zero of the filling of their IB because of too weak excitation power ranging from 5×10^{-3} to 2 mW/cm^2 for ΔEQE measurement.

We have added the PL data in the Figure 6 and Supplementary Figure 3 to support the argument, and added the following texts in pages 10 and 23, respectively, and Reference 31.

“As the excitation power density increased, the NIR emission intensity linearly increased and the emission peak wavelength blue-shifted (Fig. 6, Supplementary Fig. 3), confirming electron coupling between the PbS QDs, *i.e.*, the IB formation³¹ in the photo-absorption layers with the PbS QDs and the perovskite. The emission peak shift was nearly saturated at about 1.058 eV when excited at the excitation density of 1226 W/cm^2 . The blue-shifted energy was calculated to be 0.02 eV, which corresponds to the width of their IB.”

“which was enough low to ignore filling of the IB.”

Reply Figure 9. Excitation density dependence of NIR PL spectra of the photo-absorption layers at room temperature. The PbS volume concentrations in the photo-absorption layers were 14 volume% (A, C) and 11 volume% (B, D), respectively. Normalized PL spectra are also shown in (C) and (D). Excitation wavelength was 532 nm.

Reply Figure 10. Excitation density dependence of NIR PL emission intensity (A) and emission peak energy (B) at room temperature. Excitation wavelength was 532 nm.

(7) The overall effect on the EQE is a reduction for wavelengths below 550nm, and a small enhancement for wavelengths above 550nm. This is similar to what epitaxial ("dry method") IBSC papers report. Please provide a quantitative comparison: is the EQE enhancement bigger in IBSCs with QD or with dry epitaxial methods.

Reply:

We thank the reviewer for pointing this out. We have added the following quantitative comparison between the present solution-processed IBSCs and the previous dry-processed one in page 13 and Reference 8.

“The changes in the solar cell properties by the PbS QDs are similar to those by InAs QDs in dry-processed IBSC. However, the NIR EQE due to the PbS QDs (0.1-0.2 % at 1000 nm, Fig. 7B) was much lower than that in the dry-processed IBSC (6 % at 920 nm)⁸, suggesting that thermal excitation from the IB to the perovskite CB may be

suppressed because of the larger E_{IC} (0.8 eV) in the present IBSCs.”

(8) Smaller remark: Please rephrase the sentence on p. 5 "larger particles were formed by perturbation of electron beams". In its present form this sentence is quite unclear. Maybe what was meant that QDs coalesced/aggregated into larger QDs?

Reply:

We thank the reviewer for noticing this. As the reviewer pointed out, we have added the following text in page 5.

“aggregation of the PbS QDs with the size of 4 nm by”

Reviewers' comments:

Reviewer #1 (Remarks to the Author):

The authors have carefully addressed all the comments and the discussion section has been greatly improved.

However, I am still not sure if Type-II structure can be created when MAPbBr₃ and PbS QDs are mixed. The reference (T. Tayagaki et al., Appl. Phys. Lett., 108, 153901 (2016)) provided by the authors (Page 2, rebuttal letter) is about InP QD and thus not relevant to the topic.

The reported CBM and VBM values of I-capped PbS are ~4.6 eV and ~5.6 eV. (P. R. Brown et al., ACS Nano, 2014, 8, 5863; E. M. Miller, et al., ACS Nano, 2016, 10, 3302), however, widely varying reported CBM/VBM values are found for MAPbBr₃. Because of that, I am looking for any experimental evidence (say, UPS) to support the hypothesis of Type-II structure formation.

Reviewer #3 (Remarks to the Author):

I find the responses to my questions satisfactory. I have no further questions or reservations. I recommend the paper for publication as is.

Responses to Reviewer's Comments

Reviewer #1

The authors have carefully addressed all the comments and the discussion section has been greatly improved.

However, I am still not sure if Type-II structure can be created when MAPbBr₃ and PbS QDs are mixed. The reference (T. Tayagaki et al., Appl. Phys. Lett., 108, 153901 (2016)) provided by the authors (Page 2, rebuttal letter) is about InP QD and thus not relevant to the topic.

The reported CBM and VBM values of I-capped PbS are ~4.6 eV and ~5.6 eV. (P. R. Brown et al., ACS Nano, 2014, 8, 5863; E. M. Miller, et al., ACS Nano, 2016, 10, 3302), however, widely varying reported CBM/VBM values are found for MAPbBr₃. Because of that, I am looking for any experimental evidence (say, UPS) to support the hypothesis of Type-II structure formation.

Reply:

We thank the Reviewer for pointing this out. We agree that our present structure is not a type-II structure as schematically shown in Fig. 1(A). We believe that type-II structure can be realized by hybridizing a QD material whose VBM lies below that of MAPbBr₃ in order to increase ΔEQE and hence PCE. And also, an incorporation of a tunneling barrier with CBM above that of MAPbBr₃ into interface between the PbS QDs and

MAPbBr₃ may block the carrier transfer from the perovskite to the PbS QDs, leading to high PCE (G. Wei, et. al., *Nano Lett.* **7**, 218 (2007)). Thus, we have added the following text in page 16.

“However, an incorporation of a tunneling barrier with CB minimum above that of the perovskite into the interface between the PbS QDs and the perovskite may block the carrier transfer from the perovskite to the PbS QDs, leading to high PCE³³. And also, the carrier transfer may be suppressed by control of their energy levels like type-II structure⁴ which can be realized by hybridizing a QD material with VB maximum below that of the perovskite. In addition, the TSPA process should be enhanced by increasing the lifetime of carriers in the intermediate state with photon ratchet³⁴ and in type-II structure⁴.”

REVIEWERS' COMMENTS:

Reviewer #1 (Remarks to the Author):

The authors have addressed the concerns I had with the previous submission. I am pleased to recommend accepting the current version for publication.